# Supercluster-coupled crystal growth in metallic glass forming liquids

Yujun Xie [1,2], Sungwoo Sohn[1], Minglei Wang[1], Huolin Xin [3], Yeonwoong Jung[4], Mark D. Shattuck [5], Corey S. O'Hern [1,6,7], Jan Schroers[1] & Judy J. Cha [1,2,8]

While common growth models assume a structure-less liquid composed of atomic flow units, structural ordering has been shown in liquid metals. Here, we conduct in situ transmission electron microscopy crystallization experiments on metallic glass nanorods, and show that structural ordering strongly affects crystal growth and is controlled by nanorod thermal history. Direct visualization reveals structural ordering as densely populated small clusters in a nanorod heated from the glass state, and similar behavior is found in molecular dynamics simulations of model metallic glasses. At the same growth temperature, the asymmetry in growth rate for rods that are heated versus cooled decreases with nanorod diameter and vanishes for very small rods. We hypothesize that structural ordering enhances crystal growth, in contrast to assumptions from common growth models. The asymmetric growth rate is attributed to the difference in the degree of the structural ordering, which is pronounced in the heated glass but sparse in the cooled liquid.

[1] Department of Mechanical Engineering and Materials Science, Yale University, New Haven, CT 06511, USA. [2] Energy Sciences Institute, Yale West Campus, West Haven, CT 06516, USA. [3] Center for Functional Nanomaterials, Brookhaven National Laboratory, Upton, NY 11973, USA. [4] Nanoscience Technology Center, Department of Materials Science and Engineering, Electrical and Computer Engineering, University of Central Florida, Orlando, FL 32826, USA. [5] Department of Physics and Benjamin Levich Institute, City College of the City University of New York, New York 10031, USA. [6] Department of Physics, Yale University, New Haven, CT 06511, USA. [7] Department of Applied Physics, Yale University, New Haven, CT 06511, USA. [8] Canadian Institute for Advanced Research, Azrieli Global Scholar, Toronto, ON M5G 1M1, Canada. These authors contributed equally: Yujun Xie, Sungwoo Sohn. Correspondence and requests for materials should be addressed to J.J.C. (email: judy.cha@yale.edu)

Crystallization determines the microstructure of metals, affecting many properties, such as mechanical strength, toughness, corrosion, and electrical conductivity[1,2]. Yet, quantitative predictions of crystallization are difficult due to the complexity of the crystallization processes, which include thermodynamic and kinetic aspects that are a function of local temperature, chemistry, and their gradients[3,4]. Common crystal growth models describe growth as a phenomenon of transferring atoms at the interface between a growing solid and a structureless liquid. However, numerous studies have shown the presence of structural ordering or heterogeneities in liquids, such as the dense liquid phase[5], solute-centered quasi-equivalent clusters[6], and short-range to medium-range order[7]. Structural ordering of the liquid has been shown to affect nucleation significantly[8,9]. The mounting experimental evidence of ordering in liquids thus suggests that growth kinetics may also be affected by the local structural order of liquid if the relaxation time of such structural order is sufficiently slow. In metallic systems, the nature of structure-coupled growth is challenging to study due to very short relaxation times[10,11] and limitations of the experimental tools required to capture dynamic local atomic structures during growth at high temperature.

Here, we perform in situ transmission electron microscopy (TEM) crystallization experiments on metallic glass (MG) nanorods to determine if structure-coupled crystal growth occurs. MG nanorods are a good model system due to their slow crystallization kinetics and moderate crystallization temperatures, which are easily accessible in in situ TEM experiments to directly observe crystallization events at the atomic scale[12–14]. Our in situ TEM results suggest that crystal growth can be influenced by the presence of local structural order, which are small crystalline clusters with sizes below the critical nucleus and thus thermodynamically unstable. The main experimental finding to support this hypothesis is the observation that the growth rate of a MG nanorod undergoing crystallization upon heating is much higher than the growth rate of the same nanorod undergoing crystallization from the melt state upon cooling at the same growth temperature. This observation cannot be explained by nucleation and growth from a structureless liquid. Structure-coupled growth is further supported by manipulating the density of the small clusters through nanoscale confinement and thermal treatment of the MG nanorods. In addition, molecular dynamics (MD) simulations of binary Lennard–Jones (L–J) glasses quenched at different cooling rates show a growth rate trend that agrees with experimental observations.

## Results

**Asymmetric growth rates during isothermal crystallization.** The growth kinetics of MG nanorods were studied in situ by observing isothermal crystallization. $Pt_{57.5}Cu_{14.7}Ni_{5.3}P_{22.5}$ MG nanorods were prepared by thermoplastic forming and drop-cast onto a $SiN_x$ TEM chip for in situ experiments (Fig. 1a)[15]. Two isothermal crystallization studies were performed, one by cooling the liquid (cooled liquid) and the other by heating the glass (heated glass) (see Fig. 1b and Supplementary Fig. 1). The crystallization fronts in the two isothermal crystallization studies were tracked for a 80 nm nanorod in dark-field TEM. Figure 1c shows crystallization of the nanorod cooled from the melt (900 °C) to the isothermal crystallization temperature (420 °C) and Fig. 1d shows crystallization of the same nanorod heated from the glass (30 °C) to 420 °C (Supplementary Movies 1 and 2). Strikingly, despite the same isothermal crystallization temperature, the growth rate of the heated glass is ~25 times higher than that of the cooled liquid.

To understand the underlying mechanism of the asymmetric growth rate affected by the thermal history of the nanorod, we use aberration-corrected TEM to examine isothermal crystallization of MG rods at atomic resolution. Figure 2a shows a snapshot from a TEM movie (Supplementary Movie 3) of a 23 nm-diameter MG rod that was rapidly heated to 360 °C. After reaching the crystallization temperature, it took at least several seconds before a stable nucleus was observed. The in situ TEM movies were acquired after the nucleation event to track the crystal growth kinetics. We define the start of the movie as $t = 0$ s. The boundary between the amorphous and crystalline region is clear in this partially crystallized MG rod such that the growth rate can be measured directly. We note that the crystalline region is single-crystalline, which we have previously attributed to the lack of multiple nuclei due to the nanoscale confinement[13]. Figure 2b shows TEM snapshots separated by 1 s time intervals, tracking the growth front as a function of time. The growth front was marked as the location at which the intensity profile of the lattice fringes drops to 10% of the maximum intensity envelop (Fig. 2c, see Supplementary Note 1 and Supplementary Fig. 2 for details). Surprisingly, the measured growth rate is found to fluctuate with time (Fig. 2d) despite the constant crystallization temperature and apparent atomic plane-by-plane growth.

The fluctuations in the growth rate suggest that growth may not be solely dictated by diffusion and jumping events at the interface, in which case the growth rate would be constant in time. Variations in local atomic structures may affect the growth, possibly explaining the fluctuations. This result is in line with findings of ordering in the liquid phase[16,17], which depends on the temperature of the liquid. For example, icosahedral short-range order is reported to be more pronounced at lower temperatures and pre-existing local order can play an important role in the liquid–solid transition[18]. The size of the MG nanorods may control the number of these local structures through nanoscale confinement. Thus, we carry out isothermal experiments on three different diameter MG rods for two crystallization procedures: (1) rapid heating from the glass (30 °C) to the crystallization temperature (360 °C) and (2) rapid quench from the melt (900 °C) to 360 °C. We again observe asymmetric growth rates upon heating and cooling. Figure 3a, b show zoom-in areas of a 65 nm MG rod undergoing the two isothermal crystallizations. The average growth rate from the heated glass (Fig. 3a) is 26 times faster than that from the cooled liquid (Fig. 3b) despite the same growth temperature, in agreement with the dark field TEM result shown in Fig. 1. But here we note that the growth rates were compared for the same crystalline grain and the same crystallographic orientation (details in Supplementary Note 2). The observed single-crystalline grain takes on the C2/c structure with a chemical composition identical to that in the glass within the accuracy limit of the energy dispersive X-ray spectroscopy[13]. Figure 3c, d summarize the results for the three different-sized nanorods. We make three key observations. First, the asymmetry between the two isothermal growth rates disappears for small nanorods. Second, the average growth rate decreases with decreasing nanorod diameter for crystallization from the heated glass (red dots) while it remains constant for crystallization from the cooled liquid (blue dots) (Fig. 3d). Third, for larger rods (65 and 40 nm in Fig. 3c), the fluctuation in the growth rate is more pronounced for crystallization from the heated glass (red dots) than from the cooled liquid (blue dots). We note comparison to growth kinetics of bulk samples is difficult because bulk samples crystallize via solute partitioning dominated by chemical diffusion fields, while the nanorods crystallize into a single crystalline grain, dominated by collision-limited kinetics. There is uncertainty in measuring the location of the growth

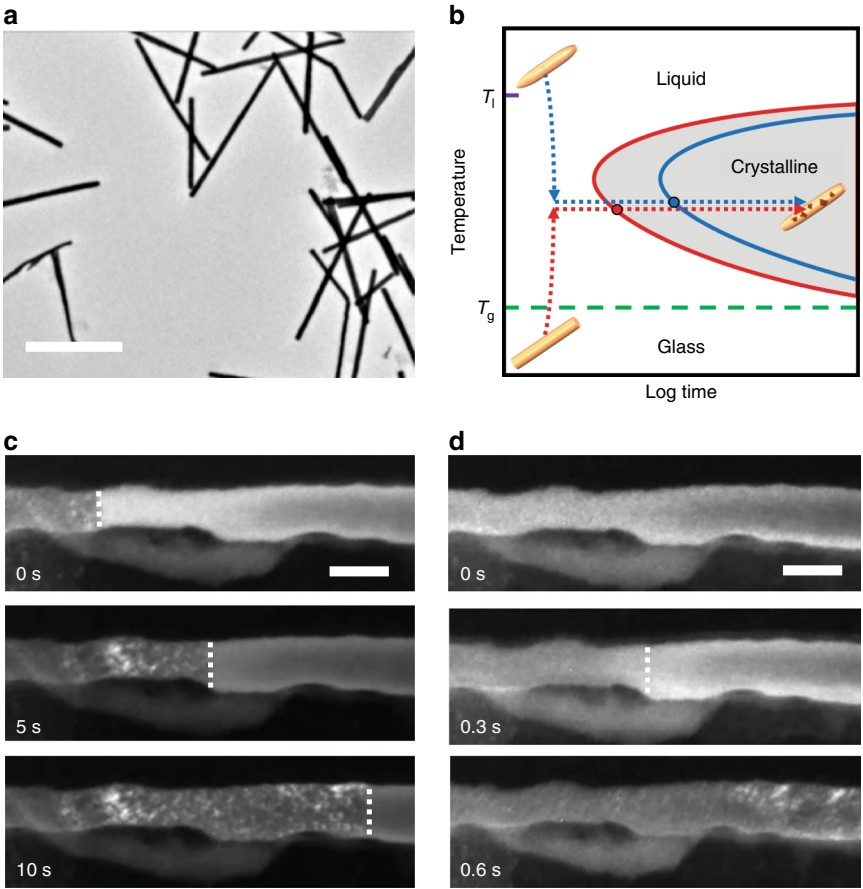

**Fig. 1** Asymmetric growth rates of MG nanorod during isothermal crystallization. **a** Bright-field TEM image of MG nanorods that are drop-cast on a thin ceramic film for in situ TEM experiments. The scale bar is 1 μm. **b** Temperature–time sequences used for the isothermal crystallization experiments. The MG nanorods were first rapidly heated to 900 °C for <5 s to remove any residual microstructures that may be left behind from thermoplastic forming. For crystallization from the melt state, the rods were then quenched and held at constant temperature for in situ observations (blue dotted line). For the studies of crystallization from the glass state, the rods were quenched to room temperature, then heated to and held at constant temperature (red dotted line). **c**, **d** Snapshot dark field (DF) TEM images of the same nanorod undergoing crystallization from the cooled liquid (**c**) and from the heated glass (**d**). The white dotted lines indicate the growth front, determined by the intensity contrast between the crystalline and amorphous regions. Despite using the same crystallization temperature of 420 °C, the growth rate from the heated glass is much higher than that from the cooled liquid. The scale bar is 80 nm

front. Since our images are resolved at the atomic scale, the estimated uncertainty is ±0.15 nm.

The asymmetric growth rates and disappearance of the growth rate asymmetry for small rods have not been observed previously. This observation is in contrast to the widely accepted explanation for the asymmetric crystallization kinetics between crystallization upon heating and upon cooling, which is attributed to the difference in the density of nuclei from the different temperature ranges at which maximum nucleation and growth rates occur[19,20], while the growth rate would be identical as long as the growth temperature is the same. In other words, crystallization of a glass during heating generates more nuclei for faster crystallization kinetics than that for crystallization of a liquid during cooling because the maximum nucleation rate occurs at lower temperatures[21]. The observed asymmetric growth rate is an additional factor to explain the asymmetric crystallization kinetics. What is the origin for the observed asymmetry in the growth rate? We hypothesize that a large number of clusters, which are smaller than critical nuclei and thus thermodynamically unstable, form during the cooling of the liquid and remain in the glass state while largely absent in the liquid state. These small clusters appear surprisingly stable kinetically to contribute to the crystal growth in the case of the heated glass, but not in the case of the cooled liquid. The vanishing asymmetry between the growth rates upon heating and cooling for small rods is

interesting. With our current hypothesis, this behavior can be attributed to nanoscale confinement: fewer clusters are available for cluster-assisted crystallization for smaller nanorods. In this case, the growth rate for the heated glass and cooled liquid do not differ significantly, since there are only few clusters in both cases.

The concept of small clusters enhancing the onset of nucleation has been already established, for example in the kinetics of a pre-treated chalcogenide glass for phase change random access memory[22–24]. These small clusters may include subcritical clusters[25], topological or chemical heterogeneities[26], thermodynamically unstable proto-nuclei[27,28], icosahedral order[18], and medium range order[29]. Here, we extend their role to growth kinetics and find that the small clusters are sufficiently stable to enhance crystal growth at elevated temperatures. Crystallization that proceeds through the coalescence of crystal embryos into the crystalline phase has actually been observed experimentally in polymer systems, such as the crystallization of o-terphenyl at temperatures at or below the glass transition temperature, with a crystallization rate that is orders of magnitude higher than expected based on homogeneous nucleation[30]. The thermal diffusivity of o-terphenyl in the liquid phase is suppressed below the glass transition temperature, which likely promoted incorporation of embryos directly into the growing phase, rather than attachment of individual o-terphenyl molecules.

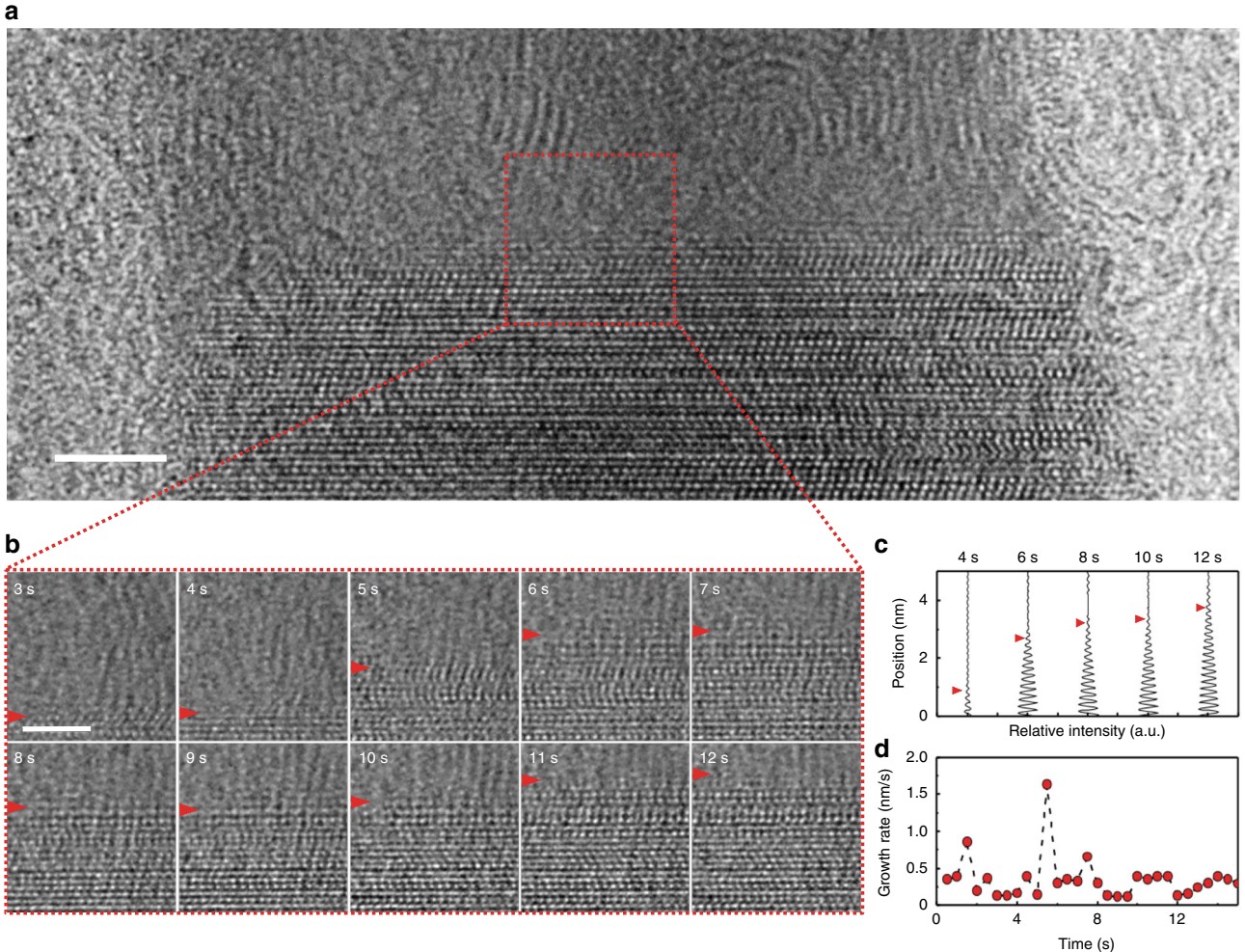

**Fig. 2** Growth rates of a 23 nm-diameter MG nanorod heated at 360 °C. **a** A TEM image from an in situ TEM movie (Supplementary Movie 3) of a 23 nm MG nanorod upon rapid heating from the glass (30 °C) to the crystallization temperature (360 °C). The boundary between the crystalline and amorphous region marks the growth front. **b** Snapshots with 1 s time intervals. The growth front (marked by the red arrow) progresses stochastically in time. **c** Intensity profiles from the filtered TEM images with red arrows that mark the position of the growth front. The growth front was marked at 10% of the maximum intensity value, allowing us to quantitatively measure the growth rate as a function of time. **d** Growth rate versus time for the 23 nm MG rod. Note that the growth rate fluctuates in time. All scale bars are 2 nm. The in situ movie was taken at the acquisition rate of 400 frames per second. The TEM images were obtained by averaging 200 consecutive frames of the TEM movie for increased signal-to-noise ratio

**MD simulations of L–J glasses**. To further test our hypothesis that transient, small clusters can couple to and enhance crystal growth, we performed MD simulations of crystallization of binary L–J glasses (Fig. 4). (The details of the simulations are provided in Methods and Supplementary Note 3.) To measure the crystal growth rate, the central region of the simulation cell was crystalline, while the outer regions were amorphous (Supplementary Fig. 3). At crystallization temperature, we observed that atomic clusters form in the amorphous region, attach to the growth front, and enhance the growth rate (Fig. 4a, b). The glass samples were prepared with varying quenching rates and heated to undergo isothermal crystallization (Supplementary Note 3). The mean growth rate, $\bar{v}$, and relative fluctuations in the growth rate, $\sigma_v/\bar{v}$, were measured as a function of the quenching rate, $R$. Glasses quenched at higher cooling rates showed slower growth rates at the same crystallization temperature (Fig. 4c). This result agrees with our hypothesis since higher quenching rates would produce fewer clusters in the glass state, leading to smaller growth rates. Thus, although binary L–J glasses do not capture the full complexity of real MGs, the results of the MD simulations qualitatively agree with those from the experiments on MG nanorods. The relatively large standard deviation of the velocity compared to the mean velocity reflects that the system size in the simulations is small (2304 atoms), in which a single event of cluster attachment will appear dramatic. The large standard deviation does not mean that the average velocity is not accurately calculated in the MD simulations, which do not suffer from the measurement uncertainties that are found in experiments. Similar cluster-coupled growth has also been observed in MD simulations of stress-driven crystallization of $Al_{50}Fe_{50}$ MGs at low temperatures (50 K)[31]. In stress-driven crystallization, ordered superclusters form in the glass matrix during stress cycles, and those that precede the growing crystal attach to the crystal, quickly advancing the growth front. The resulting intermittent growth bursts were attributed to collective hopping events of the ordered supercluster in a metabasin-to-metabasin transition due to the low thermal diffusivity of atoms. A similar mechanism can explain our observed cluster-coupled growth. Here, the diffusivity of atoms can be slowed down by nanoscale confinement or by chemical heterogeneity that generates long chemical diffusion fields, despite high temperature.

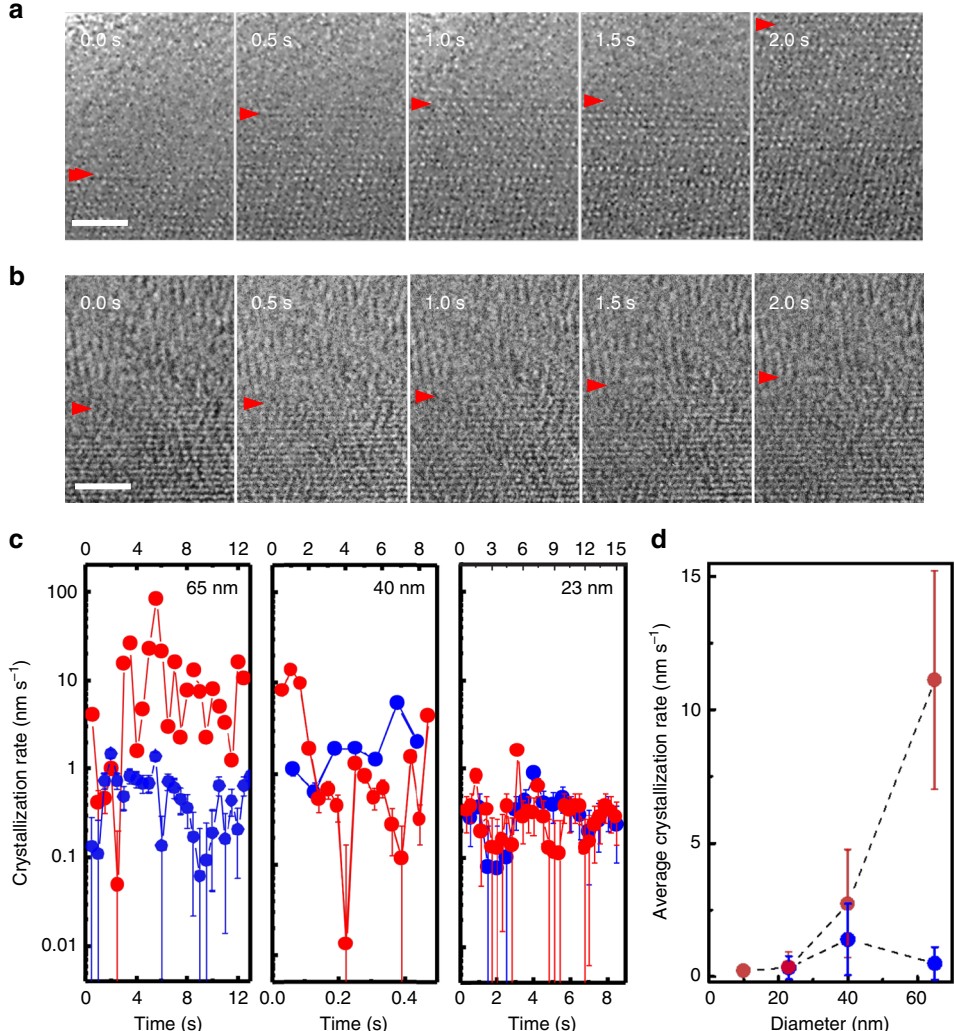

**Fig. 3** Asymmetric growth rates of MG nanorods of different diameters. **a** TEM snapshots from the in situ TEM movie of a 65 nm MG nanorod upon rapid heating from the glass (30 °C) to the crystallization temperature (360 °C). **b** TEM snapshots from the in situ TEM movie of the same 65 nm MG nanorod upon rapid quenching from the melt (900 °C) to the crystallization temperature (360 °C). Red arrows mark the growth front, measured by the intensity profiles similar to the one shown in Fig. 2c. The scale bars are 2 nm. **c** Time-resolved growth rates of nanorods of different diameters for two crystallizations: one from the heated glass (red dots, top x-axis) and the other from the cooled melt (blue dots, bottom x-axis). Three MG rods of different diameters (65 nm (left), 40 nm (middle), and 23 nm (right)) were investigated. The growth rate was measured along the perpendicular direction to the (200) crystallographic plane. The temporal resolution of the growth rate is 0.5 s except for the 40 nm rod from the melt to 360 °C. For the cooling experiment of the 40 nm rod, the growth dynamics was tracked for only a short time. In this case, the images were not averaged over 200 frames. The asymmetry between the growth rates upon heating and cooling gradually disappears with decreasing MG rod diameter. The error bars represent ±0.15 nm. **d** Average growth rate as a function of the MG rod diameter. The vanishing difference in the growth rate with decreasing rod diameter is clear. The error bars represent the standard deviation of the growth rate

We also consider other possibilities to explain the asymmetry and fluctuations in the growth rate. For example, the crystallization rate can be affected by the fictive temperature. Generally, the fictive temperature is lower for slowly cooled liquids, which gives rise to a higher thermodynamic driving force for faster crystallization. However, the effects of the fictive temperature are only present near or below glass temperature. In our experiments, because the rods are crystallized at a temperature that is significantly higher than the glass temperature, the effects of the fictive temperature are negligible. Another possibility is deviation in thermal transport of MG rods at the nanoscale, particularly making thermal transport inefficient for smaller rods. However, in this case, we would expect larger fluctuations and larger asymmetry in the growth rate for smaller rods, which is the opposite of what we observe. In addition,

the growth rates for crystallization from the cooled liquid would also show size-dependent effects if thermal transport at the nanoscale were the cause for the asymmetry. The observed fluctuations could also arise from surface nucleation in which additional atomic layers randomly attach to the growing solid. However, in this case, the growth fluctuations would also be present for crystallization during cooling, which we do not observe. Moreover, the probabilistic nature of surface nucleation does not explain the asymmetry in growth rate. We also note that the observed size dependence and fluctuations in the growth rate may be related to the potential lack of stress relaxation during growth from the glassy state, as previously studied in a polymer system[32]. Other extrinsic factors that could affect crystallization, such as electron beam irradiation effects, oxidation effects, or carbon build-up, have been

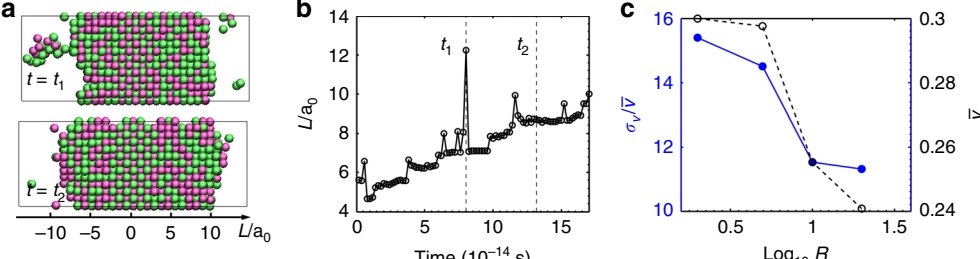

**Fig. 4** MD simulations of isothermal crystallization of binary Lennard–Jones glasses. **a** Two snapshots of a Lennard–Jones binary glass sample during an isothermal simulation at $T = 0.1$ at two times $t_1$ (top) and $t_2$ (bottom). $t_1$ and $t_2$ are defined in **b**. The snapshots only show crystalline atoms for clarity. The full configuration (crystalline and amorphous atoms) are in Supplementary Fig. 3. The pink and green colors represent the A and B atom types, respectively. **b** Location of the left boundary, $L$, between the crystalline region and the remaining glass region as a function of time $t$ for simulations in (**c**). The boundary location is normalized by the size of the large atom ($\sigma_A$). The boundary location increases with time, indicating crystal growth. Occasionally, the boundary location exhibits large fluctuations and sudden jumps, such as the one at $t_1$. The large peak at $t_1$ is due to a cluster that is connected to the crystalline section in the middle. **c** (Left) Standard deviation of the growth rate normalized by the mean growth rate, $\sigma_v/\bar{v}$, and (right) the mean growth rate, $\bar{v}$, as a function of the cooling rate, $R$, used to generate the glasses. Sixteen independent simulation samples were run to obtain the standard deviation and mean of the growth rate. Glasses prepared at slower cooling rates exhibit larger standard deviations in the growth rate and higher average growth rates

considered and ruled out in our previous in situ TEM experiments (Supplementary Note 4)[13].

**Presence of small, ordered clusters in heated glass state**. The presence of thermodynamically unstable clusters that persist long enough to couple to crystal growth is surprising. The lifetime of clusters in a simple polymorphic system can be approximated within classical nucleation theory by the temperature-dependent transient time[33]. Assuming classical nucleation theory, for the Pt-based MGs we study, calculated transient times at 340 °C are ~0.128 s [34,35] (Supplementary Note 5). However, crystallization occurs on the order of one hundred seconds. Hence, the thermal history of the cluster distribution would be erased on this time scale, resulting in identical growth rates for both the heated glass and the cooled liquid during crystallization. We directly observe that the lifetime of a ~2 nm cluster at 340 °C is at least ~10 s (Fig. 5a and Supplementary Movie 4). Clearly, assuming a polymorphic transition that requires only rapid topological fluctuations is an oversimplification for this Pt-based MG. As the composition of the crystalline phase is generally different from the liquid composition, an inherent feature of glass forming melts, chemical fluctuations are required during the nucleation process. These fluctuations are orders of magnitude slower than topological fluctuations. Whereas topological fluctuations occur on the atomic scale, chemical fluctuations require diffusion over large lengthscales[36]. These slow processes may explain the observed long lifetime of the small clusters that enhance crystal growth.

If small clusters can couple to crystal growth, controlling the population and stability of such small clusters can drastically change the growth dynamics. To test this, we applied a different heating profile to tune the cluster population. Instead of rapidly reaching the crystallization temperature from the glass state, we gradually heated a 35 nm-diameter MG nanorod from the glass state to 340 °C. A snapshot of the partially crystallized rod is shown in Fig. 5b, where the boundary between the amorphous and crystalline regions is marked by white arrows. Densely populated, ordered clusters can be seen in the amorphous region. Unlike the crystalline region, the small clusters do not show apparent structure patterns in corresponding diffractograms (Fig. 5b insets). The clusters are quite stable, persisting for several tens of seconds at 340 °C, supporting the hypothesis that thermodynamically unstable clusters can have long structural relaxation times and couple to growth (Supplementary Movie 5). Some of the small clusters we observe may be electron-beam

induced. However, beam-induced cluster formation cannot explain the observed asymmetry in the growth rates because the beam effects would equally apply to crystallization during heating and cooling.

The hypothesis of cluster-coupled growth is summarized in the schematics in Fig. 5c, d. The top schematic of Fig. 5c illustrates the classical theory used to explain asymmetric crystallization kinetics. The bottom schematic of Fig. 5c illustrates our modification to the growth part of the classical theory. A key difference from the classical theory is the role of small clusters during growth: small clusters can directly couple to growth instead of forming nuclei, which would increase the growth rate for crystallization from the glass significantly (Fig. 5c, bottom). For crystallization from the cooled liquid, cluster formation is expected to be negligible at the crystallization temperature. Indeed, the growth rates by cooling the melt state of the three different MG rods were comparable and growth rate fluctuations were minimal (Fig. 3c, d). For crystallization from the glass state, small clusters can form during quenching of the melt and persist long enough to couple to growth at the nanoscale. Figure 5d thus introduces our cluster-coupled growth as an additional growth mechanism to the existing common growth model and particle attachment.

The origin of the observed long stability of small clusters in MG rods may be due to sluggish chemical fluctuations required for formation and decay of clusters[37,38]. The question of why the clusters enhance, rather than impede, the growth, remains to be answered. The local ordering of the supercooled liquid has been used to explain the fast growth kinetics of pure metals. For crystallization of pure metals from their melts, recent MD simulations found that the liquid near the liquid/solid interface has crystalline ground states, which effectively leads to the absence of a barrier to crystallization for ultrafast crystal growth[39]. We have also observed suppression of the growth rate due to a nearby crystal that was misaligned with the growing crystal[14]. The key factor in determining whether clusters enhance or impede growth may lie in the difference between the thermodynamic stability and size of the stable crystal versus metastable clusters. The presence of clusters smaller than critical nuclei in the glass state is known for many materials, such as polymers[40], chalcogenide glasses[11], and biominerals[41]. Crystallization enhancement due to these clusters was generally attributed to the enhancement of nucleation rather than growth. Here, we show that growth can also be enhanced by the presence of clusters. The proposed cluster-coupled growth is different from Ostwald ripening and particle attachment[3], which

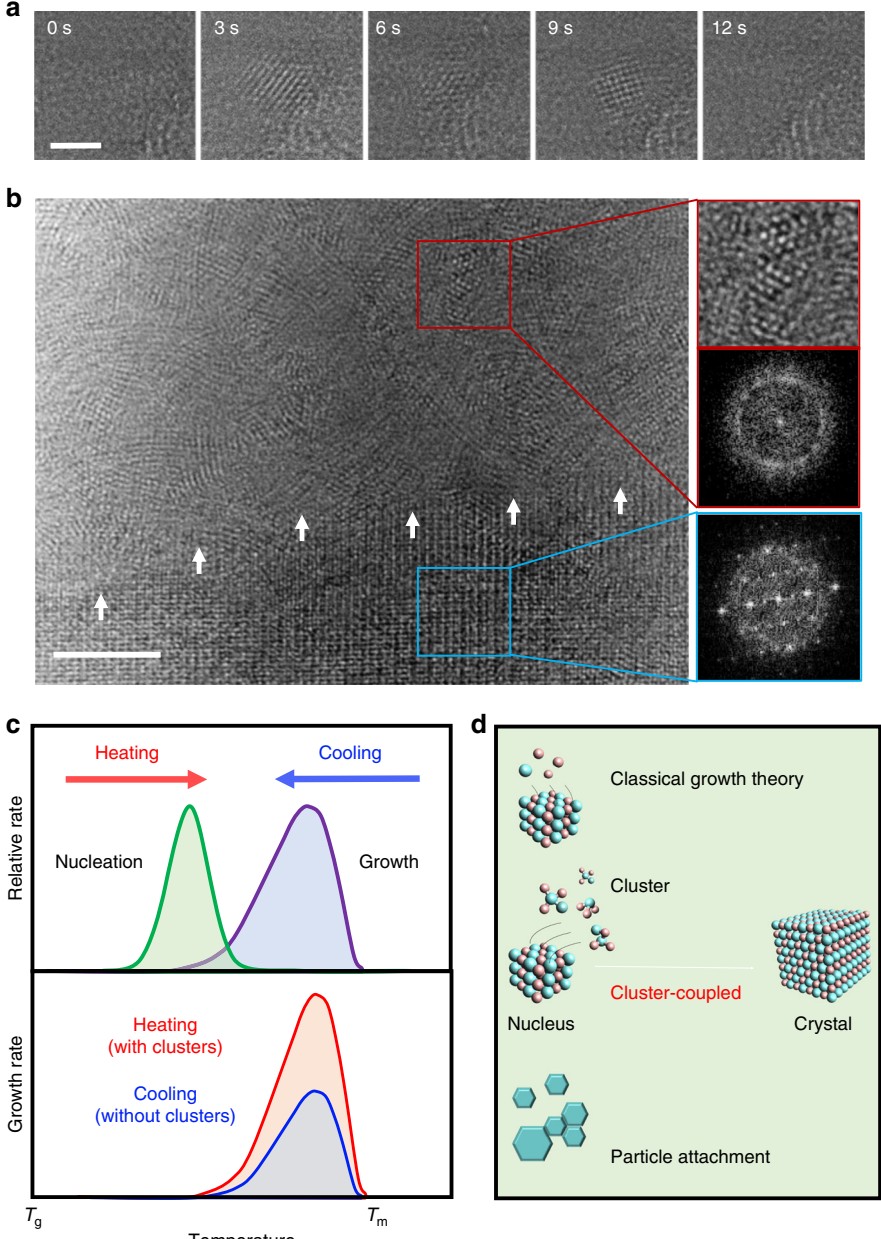

**Fig. 5** Presence of small clusters and the proposed cluster-coupled growth mechanism. **a** TEM images from isothermal crystallization (Supplementary Movie 4) showing a cluster in a 20 nm MG nanorod at 340 °C. A cluster forms, rotates, and disappears within 6 s at 340 °C. The scale bar is 2 nm. **b** A zoom-in area of a 35 nm-diameter MG nanorod during isothermal crystallization from the glass state at 340 °C (Supplementary Movie 5). Clusters with widths around 2 nm exist in the amorphous region (cropped image in red box) and persist for several tens of seconds despite the high temperature of 340 °C. The scale bar is 3 nm. **c** Schematics that describe the classical theory of crystal growth (top) and a modified description of crystal growth in which small clusters can enhance the growth (bottom). **d** Schematic illustration showing different growth pathways, including the proposed cluster-coupled growth

require the presence of an interfacial boundary. In cluster-coupled growth, the small clusters are dynamical structures in the matrix that fluctuate in and out of the liquid phase. The present results demonstrate that classical growth models are inadequate for describing crystallization of most metallic alloys.

## Methods

**Preparation of MG nanorods**. The synthesis of $Pt_{57.5}Cu_{14.7}Ni_{5.3}P_{22.5}$ MG nanorods is reported in detail in our previous paper[12]. A high purity master alloy of ~20 g with nominal compositions was melted in a vacuum-sealed quartz tube and fluxed with dehydrated boron trioxide ($B_2O_3$, ~10 g) for 30 min at 1200 °C to remove impurities, 450 °C above the liquidus temperature of $Pt_{57.5}Cu_{14.7}Ni_{5.3}P_{22.5}$. The fluxed alloy was re-melted at 1100 °C for 2 min and quenched with water. To fabricate the nanorods, a piece of the bulk alloy was positioned on a

commercially available anodized aluminum oxide (AAO, Synkera Inc.) with the pore size ranging from 13 to 200 nm in diameter. By pressing the AAO mold against the bulk alloy under a load with linear ramping from 0 to 100 kN in 2 min at 260 °C, nanorods were thermoplastically formed. To detach the nanorods, the AAO mold was dissolved in a 20 wt% potassium hydroxide (KOH) solution at 80 °C for 10 h and repeatedly rinsed using distilled water and isopropanol (IPA). The nanorods were collected by detaching them from the MG plate via sonication.

**In situ TEM experimental details**. In situ aberration-corrected TEM experiments were carried out at Brookhaven National Laboratory with the FEI Titan 80-300 operating at 300 keV. Dark field in situ TEM experiments were carried out using the FEI Tecnai Osiris 200 keV at Yale. Nanorods dispersed in IPA were drop casted on an in situ thermal chip for crystallization experiments using a heating holder (Aduro 300DT System by Protochips Inc.). The in situ thermal chips consist of a

holey silicon nitride ($Si_3N_4$) with a thin amorphous carbon film overlay and metal electrodes for Joule heating. In situ TEM experiments were recorded using a Gatan K2-IS camera. To avoid side effects in the crystallization experiments from the thermoplastic forming process, the MG nanorods were first heated to 900 °C, ~300 °C higher than the liquidus temperature, for no more than 5 s. For the isothermal heating experiments, we rapidly quenched the nanorods from 900 °C to room temperature, and heated the nanorods to the crystallization temperature with a maximum ramping rate of ~$10^6$ °C $s^{-1}$. For the isothermal cooling experiments, we rapidly quenched the nanorods from 900 °C to the crystallization temperature with a maximum ramping rate of ~$10^6$ °C $s^{-1}$. In both cases, we visualized the crystal growth in real time. The temperature of the nanorods was assumed to be the same as the temperature of the in situ TEM thermal grids, whose temperature was read out by the power controlling system.

MD simulations of crystallization of binary L–J glasses: We performed MD simulations of binary L–J mixtures to investigate crystal growth kinetics and compare to the experimental results. The MD simulations used periodic boundary conditions in all three spatial dimensions. Thus, the results from the MD simulations more closely mimic bulk samples, rather than those with nanoscale confinement. We simulate 2304 atoms; half of the atoms are small and the other half are large with diameter ratio $\sigma_A/\sigma_B = 1.02$, which allows us to study both amorphous and crystallized samples[42]. The pairwise interaction potential between atoms $i$ and $j$ is described as follows:

$$u\left(r_{ij}\right) = 4\varepsilon\left[\left(\frac{\sigma_{ij}}{r_{ij}}\right)^{12} - \left(\frac{\sigma_{ij}}{r_{ij}}\right)^6\right] \qquad (1)$$

where $r_{ij}$ is the center-to-center distance between atoms $i$ and $j$, $\varepsilon$ is the depth of the attractive part of the interaction, and $\sigma_{ij} = (\sigma_i + \sigma_j)/2$ is the average diameter. $u(r_{ij})$ has been truncated and shifted so that the interatomic potential energy and force vanish for separations $r_{ij} > 2.5\sigma_{ij}$. Periodic boundary conditions were applied in all three dimensions. The sample lengths are 25.72, 9.64, and 9.64 in the $x$-direction, $y$-direction, and $z$-direction, respectively, in units of the small atom diameter $\sigma_B$. We implemented a Gaussian constraint thermostat to maintain constant temperature. Physical quantities from the simulations are presented in units of $\varepsilon$, $\sigma_B$, $\sigma_B\sqrt{m/\varepsilon}$, and $\varepsilon/k_B$, for energy, length, time, and temperature scales, respectively, where $m$ is the mass of both the large and small atoms.

## Data availability

All raw data presented in this work are available from the corresponding authors upon request.

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

## Acknowledgements

Microscopy facilities used in this work were supported by the Yale Institute for Nanoscience and Quantum Engineering (YINQE). This research used resources of

the Center for Functional Nanomaterials, which is a U.S. DOE Office of Science Facility, at Brookhaven National Laboratory under Contract No. DE-SC0012704. J.C. acknowledges support from CIFAR Global Scholars. Y.X. acknowledges support from NSF EFMA 1542815. M.W. and C.S.O. acknowledge support from NSF MRSEC DMR 1119826.

## Author contributions

Y.X. and S.S. equally contributed to this work. Y.X., S.S. and Y.J. carried out the in situ TEM experiments under the direction of J.S. and J.J.C. and with help from H.X. S.S. performed the nanomolding experiments under the direction of J.S. M.W., M.D.S., and C.S.O. carried out the MD simulations. S.S., Y.X., J.S. and J.J.C. analyzed the TEM data. All authors contributed to the discussion on the results and writing of the manuscript.

## Additional information

**Competing interests:** The authors declare no competing interests.

