## [Peer Review File · Nature Communications]

Reviewers' comments:

Reviewer #1 (Remarks to the Author):

This is an interesting paper. Using in situ as well as high-resolution and Cs-corrected TEM, the authors directly track the growth of an already nucleated crystal. They demonstrate that at the same crystallization temperature, the crystal growth rate in a heated metallic glass nanorod (23 to 80 nm in diameter) is some 25 x higher than that of a cooled metallic liquid rod of the same size. This asymmetry has an origin different from the known mechanism: previously an asymmetry like this was attributed to the enhancement of nucleation rather than growth. There it is the difference in the density of nuclei, due to the different temperatures at which the nucleation rate and the growth rates peak [their 21,22]. Here, the growth is enhanced by the presence of clusters, i.e., ordering of the glass structure in the sample heated to the crystallization temperature.

I recommend that the authors consider the following in their revision.

1. "Cluster-Coupled" in the title is a bit confusing. The entities they observe that fluctuate in and out seem to be "superclusters" at or beyond medium-range level. If you simply state "clusters", their sizes can range over a wide range, including sub-nm ones all the way down to short-range nearest neighbor polyhedra level. The latter SRO is widely available in the glass structure, but has not yet reached the level of structural order that is coupling to the growing crystal discussed here.

Instead, these entities could be subcritical nuclei of the final crystallization product, or embryos of some other metastable crystal state, popping out with a lifetime of seconds in front of the advancing crystal front.

2. That these precursors become part of the growing crystal, enhancing its growth rate, is an interesting process. The authors report fluctuating rate at the growth front. I would like to bring to their attention a recent modeling paper, Yunwei Mao et al., Phys. Rev. B 91, 214103 (2015). The MD model there contains an analysis, more in depth than the current L-J model MD simulation, and actually talked about how patches of ordered structure in a metallic glass could join the crystal and cause intermittent advancement of the front, spatially and temporally. It was for stress-activated crystallization process, rather than thermally induced here, but the idea may be the same. It explains the jerky and enhanced growth that appears to be a surprise at the first glance (too fast, or not even possible).

3. Ordered precursors promote interface forward movement, but also slow down dynamics. So apparently the former overpowers the latter in this case, perhaps because adding ordered patches is more efficient than hopping of individual atoms to attach?

4. Could the authors discuss a bit more what the crystal phase is, how is the interface growth rate compared with other reports before, and how does it compare with previous models of crystal growth from the melt (such as the ones Mike Aziz and David Turnbull use)? Here we have chemical partitioning, so the interface rate should indeed be way slower than the collision limited case of an atomic system. But some clues may be found in solidification papers, and in AL Greer's review article of previous cases of amorphous alloys (crystal growth speed in chalcogenides and metallic systems).

Reviewer #2 (Remarks to the Author):

The authors present the experimental measurement of crystal growth rates for a Pt-P based metallic glass, reporting that crystal growth rates measured over a period of seconds is an order of magnitude faster if heated from the glass state than if cooled from the liquid. The authors note that the growth rate of the heated sample converges to that of the cooled sample as the rod diameter decreases. Large fluctuations in the growth rate are also noted for the larger samples. The conclusion is that crystal growth from the heated sample is being assisted by the presence of crystal-like clusters. Simulation data is presented to support this conclusion.

While the observations are interesting (and I fully concur with the authors' criticism of the classical growth models), I find the evidence presented for cluster involvement weak and the proposed idea has enough holes in it to render it unlikely. As a result I cannot recommend publication of this paper.

The authors suggest that, despite the fact the clusters formed with an orientation completely uncorrelated with that of the growing crystal, they can still assist the growth of the crystal rather than obstructing it. This idea has been suggested before. In 1995 Oguni [PRB 52, 3900 (1995)] suggested something similar to explain fast crystal growth at and below T_g . I certainly recommend the literature on this problem to the authors. Current thinking on the cause of fast crystal growth from the glass [see Ediger and Yu, JPCB 119, 10124 (2015)] has focused on the failure of effective stress relaxation during the growth from the glassy state - an explanation that would make better connection with the observed size dependence and the large fluctuations observed by the authors.

The simulation results - that crystals grow slower when quenched fast than when quenched slow - really do not provide any strong support for the proposed cluster model. The simpler explanation is that the fictive temperature of the rapidly cooled liquid is higher than that of the more slowly cooled liquid and so the thermodynamic driving force for crystallization is less (another alternative explanation of the experimental observation).

Some general comments:

- i) Error bars should be included on the plot of the average crystallization rate in Fig. 3c.
- ii) It appears that some references are missing - my copy only goes up to citation 37 and the paper cites papers up to 44.

Reviewer #3 (Remarks to the Author):

In this work, the authors investigated the crystal growth kinetics of Pt-based metallic glass nanorods using in situ TEM measurements. Specifically, the nanorods were first melted, then either rapidly quenched to crystallization temperature T_c , or to a much lower temperature and re-heated to T_c . The materials were then kept at T_c , at which isothermal crystallization occurred and the growth rate was recorded via in situ TEM. It was found that the re-heated material had crystal growth rate 26 times faster than the direct cooled sample. The authors attributed the difference to the direct attachment of metastable nanoclusters below size of critical nuclei, and supported the claim by comparison with MD simulation of a binary Lennard-Jones system.

The carefully conducted in situ TEM measurements and the subsequent analysis convincingly demonstrated the difference in crystal growth speed from the liquid state versus the glass state, at least for the 65nm nanorods. The hypothesized cluster-coupled mechanism requires better scrutiny on the applicability/limitations for reasons listed below.

First, the authors presented their results in contrast to established theory where cooled sample contains more nuclei, which in turn will have higher nucleation rate but the same growth rate.

However, the authors had identified that due to nano-confinement, only one nucleus grew (and formed a single crystal). By comparing the time scale involved: tens of seconds for the growth

versus sub-microsecond for nucleation, the crystallization process of this material appears growth-

limited. Without comparing with bulk material (for which the classic theory is based on) and analyze the density of nuclei, we cannot rule out with confidence that what's observed is merely a manifestation of an evolving super-critical nucleus engulfing sub-critical clusters. If one should expect multiple super-critical nuclei to form in 25 and 40 nm rods during the very rapid heating, then the argument of that we are observing something different from classic theory will be more convincing. As of now, it can be argued that the observation is size/material-system specific instead of having wide range implications.

Following the above argument, the appearing lack of difference in 25/40 nm material should be addressed. While the rate of 40 nm heated material as shown in fig 3(d) does appear higher, no error bar is given. Based on figure 3(c), the rate of cooled versus heated material seem very close factoring in fluctuations.

Also, the authors have acknowledged that any size/population difference in sub-critical nuclei distribution should affect the rate. The authors claimed to test this (line 193), but only went on to address the appearance of ordered region instead of comparing growth speed.

Finally, the MD simulations showed cluster attachment can occur and will create significant spikes in growth rate curve. The simulation setup is for growth of a single crystal in a confined geometry, so again we have the problem of whether we are just observing size effect instead of new physics. The simulation also claimed that material that underwent faster cooling also crystallized faster. This, however, is not addressed in the experiment. Also figure 4(g) is a bit confusing. The left-hand axis plotting normalized standard deviations shows values of ~11 to 15. If it implies that the standard deviations are ~15 times the size of average speed, then any "difference" in the speed (about 20%) is meaningless.

Based on issues raised above, the authors should revise the manuscript before it can be published in high-impact journal such as Nature Communications.

We thank the reviewers for their constructive and insightful comments, which have helped us to improve the revised manuscript. We provide a point-by-point response to their comments below. For clarity, the reviewers' comments appear in black and our responses appear in blue.

Reviewer #1 (Remarks to the Author):

This is an interesting paper. Using in situ as well as high-resolution and Cs-corrected TEM, the authors directly track the growth of an already nucleated crystal. They demonstrate that at the same crystallization temperature, the crystal growth rate in a heated metallic glass nanorod (23 to 80 nm in diameter) is some 25 x higher than that of a cooled metallic liquid rod of the same size. This asymmetry has an origin different from the known mechanism: previously an asymmetry like this was attributed to the enhancement of nucleation rather than growth. There it is the difference in the density of nuclei, due to the different temperatures at which the nucleation rate and the growth rates peak [their 21,22]. Here, the growth is enhanced by the presence of clusters, i.e., ordering of the glass structure in the sample heated to the crystallization temperature.

I recommend that the authors consider the following in their revision.

1. "Cluster-Coupled" in the title is a bit confusing. The entities they observe that fluctuate in and out seem to be "superclusters" at or beyond medium-range level. If you simply state "clusters", their sizes can range over a wide range, including sub-nm ones all the way down to short-range nearest neighbor polyhedra level. The latter SRO is widely available in the glass structure, but has not yet reached the level of structural order that is coupling to the growing crystal discussed here. Instead, these entities could be subcritical nuclei of the final crystallization product, or embryos of some other metastable crystal state, popping out with a lifetime of seconds in front of the advancing crystal front.

We agree that the current title can be misleading. We do not believe that the clusters we observe are short range order (SRO) structures. Currently, the experimental methods do not have sufficient spatial resolution to definitively address this issue. For example, the clusters could be subcritical nuclei of the final crystalline phase or embryos of other metastable crystalline phases. Regardless, these clusters are metastable and smaller than stable nuclei.

In response, we have changed the title in the revised manuscript to 'Supercluster-coupled crystal growth in metallic glass forming liquids.'

2. That these precursors become part of the growing crystal, enhancing its growth rate, is an interesting process. The authors report fluctuating rate at the growth front. I would like to bring to their attention a recent modeling paper, Yunwei Mao et al., Phys. Rev. B 91, 214103 (2015). The MD model there contains an analysis, more in depth than the current L-J model MD simulation, and actually talked about how patches of ordered structure in a metallic glass could join the crystal and cause intermittent advancement of the front, spatially and temporally. It was for stress-activated crystallization process, rather than thermally induced here, but the idea may be the same. It explains the jerky and enhanced growth that appears to be a surprise at the first glance (too fast, or not even possible).

We thank the reviewer for pointing out this paper (PRB 91, 214103 (2015)). Although the origin of crystallization is different (stress- versus thermally activated), the PRB paper describes in detail how ordered clusters can attach to the growing crystal, leading to intermittent advancement of the interface between the amorphous and crystalline regions. As the reviewer emphasizes, the explanation presented in the PRB paper can also be applied to our results.

In response, we added the paper mentioned by the reviewer (PRB 91, 214103 (2015)) as Ref. [31] and the following discussion (on page 8) in the revised manuscript.

"Similar cluster-coupled growth has also been observed in MD simulations of stress-driven crystallization of $\text{Al}_{50}\text{Fe}_{50}$ metallic glasses at low temperatures (50 K)³¹. In stress-driven crystallization, ordered superclusters form in the glass matrix during stress cycles, and those that precede the growing crystal attach to the crystal, quickly advancing the growth front. The resulting intermittent growth bursts were

attributed to collective hopping events of the ordered supercluster in a metabasin-to-metabasin transition due to the low thermal diffusivity of atoms. A similar mechanism can explain our observed cluster-coupled growth. Here, the diffusivity of atoms can be slowed down by nanoscale confinement or by chemical heterogeneity that generates long chemical diffusion fields, despite high temperature.”

3. Ordered precursors promote interface forward movement, but also slow down dynamics. So apparently the former overpowers the latter in this case, perhaps because adding ordered patches is more efficient than hopping of individual atoms to attach?

The reason the ordered precursors promote, rather than suppress, the advancement of the growth front is unclear. We have actually observed the ordered crystallite slowing down the growth of a larger crystal nearby, which was reported in JOM 69, p.2187-2191 (2017) (cited as Ref. [16] in the manuscript). We present Figure 3 of the 2017 JOM paper below.

Reprinted by permission from Springer Nature: Xie, Y., Sohn, S., Schroers, J. & Cha, J. Direct Observation Through In Situ Transmission Electron Microscope of Early States of Crystallization in Nanoscale Metallic Glasses. JOM 69, 2187-2191 (2017). <https://link.springer.com/article/10.1007/s11837-017-2579-0>

Here, the key difference is the size of the crystallite. In the JOM paper, the preceding crystallite is quite large, ~ 40 nm long and 17 nm wide. Thus, it makes sense that this crystallite will slow the growth of the other grain until the crystallographic orientation of the two aligns. In the current manuscript, the clusters that attach to the growing crystal are subcritical nuclei or superclusters whose structural ordering may not be as long-ranged. As the reviewer hints, perhaps adding ordered patches is more efficient than the attachment of individual atoms. This was the main mechanism to explain the intermittent growth bursts observed in the MD simulations of stress-induced crystallization (PRB 91, 214103 (2015)). Despite the elevated temperature of 360°C in our case, the diffusivity of individual atoms can be low either due to nanoscale confinement or due to chemical heterogeneity. This point was discussed in response to Question #2.

In response to this comment, we have added the following sentences (page 11) to the revised manuscript.

“[The question of why the clusters enhance, rather than impede, the growth, remains to be answered.] We have observed suppression of the growth rate due to a nearby crystal that was misaligned with the growing crystal¹⁶. The key factor in determining whether clusters enhance or impede growth may lie in the difference between the thermodynamic stability and size of the stable crystal versus metastable clusters.”

4. Could the authors discuss a bit more what the crystal phase is, how is the interface growth rate compared with other reports before, and how does it compare with previous models of crystal growth from the melt (such as the ones Mike Aziz and David Turnbull use)? Here we have chemical partitioning, so the interface rate should indeed be way slower than the collision limited case of an atomic system. But some clues may be found in solidification papers, and in AL Greer’s review article of previous cases of amorphous alloys (crystal growth speed in chalcogenides and metallic systems).

Crystal phase. Two known competing crystalline products for the Pt-based BMG are P_2Pt_5 with monoclinic C2/c structure and CuP_2 with P21/c structure. Selected area electron diffraction patterns obtained from the crystalline grain suggest C2/c structure based on the symmetry of the diffraction patterns. We have examined this phase more closely in our previous paper (Nat. Communications 8:1980 (2017), Supplementary Note 3, cited as Ref. [15] in the manuscript).

Interface growth rate. Comparing our growth rate to previous reports from the melt is tricky due to nanoscale confinement. In our first study (Nat. Communications 6:8157 (2015), cited as Ref. [14]), we report that the crystallization rate is greatly suppressed for nanorods with diameters below ~ 50 nm at the temperature ramping rate of 0.67°C/s , compared to bulk studies. The suppressed crystallization kinetics was reproduced as lower critical heating and cooling rates in the follow-up study (Nat. Communications 8:1980 (2017), cited as Ref. [15]). These observations were attributed to apparent high viscosity and low nucleation rate induced by the nanoscale confinement. In addition, nanoscale confinement leads to a single-crystalline grain with a chemical composition identical to the glass composition rather than solute partitioning. As the reviewer points out, the long-range diffusion-limited solute partitioning will be much slower than the collision-limited single grain growth. In addition, most growth studies from the melt are *ex situ* and post-mortem, rendering direct comparisons difficult. Due to these reasons, although we appreciate the reviewer's suggestion and have considered it carefully, we feel that a comparison between our growth kinetics to bulk studies is not necessary.

In response to the reviewer's comment, we added the following statement on page 5: "The observed single-crystalline grain takes on the C2/c structure with a chemical composition identical to that in the glass within the accuracy limit of the energy dispersive X-ray spectroscopy¹⁵."

We also added the following comment on page 6.

"Comparison to growth kinetics of bulk samples is difficult because bulk samples crystallize via solute partitioning dominated by chemical diffusion fields, while the nanorods crystallize into a single crystalline grain, dominated by collision-limited kinetics."

Reviewer #2 (Remarks to the Author):

The authors present the experimental measurement of crystal growth rates for a Pt-P based metallic glass, reporting that crystal growth rates measured over a period of seconds is an order of magnitude faster if heated from the glass state than if cooled from the liquid. The authors note that the growth rate of the heated sample converges to that of the cooled sample as the rod diameter decreases. Large fluctuations in the growth rate are also noted for the larger samples. The conclusion is that crystal growth from the heated sample is being assisted by the presence of crystal-like clusters. Simulation data is presented to support this conclusion.

While the observations are interesting (and I fully concur with the authors' criticism of the classical growth models), I find the evidence presented for cluster involvement weak and the proposed idea has enough holes in it to render it unlikely. As a result I cannot recommend publication of this paper.

The authors suggest that, despite the fact the clusters formed with an orientation completely uncorrelated with that of the growing crystal, they can still assist the growth of the crystal rather than obstructing it. This idea has been suggested before. In 1995 Oguni [PRB 52, 3900 (1995)] suggested something similar to explain fast crystal growth at and below T_g . I certainly recommend the literature on this problem to the authors.

We thank the reviewer for mentioning PRB 52, 3900 (1995), which describes crystallization of *o*-terphenyl, aromatic hydrocarbon chains. The central finding of the paper is that 'crystallization proceeds through the coalescence of crystal embryos into the crystalline phase on the liquid-crystal interface,' which lowers the 'effective interfacial energy of the embryo due to the coalescence.' The measured growth rate was orders of magnitude higher than the growth rate calculated by assuming homogeneous nucleation. The

presence of embryos was interpreted to be consistent with the residual entropy left in the system. The authors note at the end similar crystallization behavior 'would be observed in the future in many systems of fragile liquids.' Figure 8 of PRB 52, 3900 (1995) shows that the observed growth rate occurs 40 degrees Celsius lower than the previous study (Experimental Thermodynamics, 1, p.133 (1968)).

This paper is relevant to our work and we cite it as Ref. [30] in the revised manuscript. We still strongly believe that our study is not a mere repeat or rehash of what is already known, but adds critical value to the field for several reasons. First, because the system we consider is metallic, not polymeric, one would naively expect much faster diffusion kinetics and thermal instability for metastable embryos. Thus, it is not obvious that such cluster-coupled growth should occur for metallic glasses heated at the crystallization temperature. Second, the studies in PRB 52, 3900 (1995) induce crystallization at or below T_g , a regime where thermal diffusion is expected to be low, while our experiments were conducted above T_g (T_g of the bulk crystal is $\sim 230^\circ\text{C}$, while the crystallization experiments were conducted at 360°C). Thus, the mechanism of collective hopping events of an embryo attaching to the growing surface rather than collision-limited single atom attachment is even more surprising. Finally, we investigated two different processing paths, heating the glassy state versus cooling the liquid state, and observed a significant difference in the growth rates, which was not discussed in the previous paper. This difference in growth rates is a major finding in our work and has never been reported before.

In summary, as the reviewer points out, crystallization via embryo attachment has been observed previously in polymers. However, the fact that it occurred in metallic systems at high temperatures above T_g is new and unexpected. The cluster-attachment mechanism observed in metallic systems is attributed to the low diffusivity of atoms from nanoscale confinement and chemical heterogeneity.

In response, we cite the previous work (PRB 52, 3900 (1995)) as Ref. [30] and added the following discussion on page 7 in the revised manuscript:

"Crystallization that proceeds through the coalescence of crystal embryos into the crystalline phase has actually been observed experimentally in polymer systems, such as the crystallization of o-terphenyl at temperatures at or below the glass transition temperature, with a crystallization rate that is orders of magnitude higher than expected based on homogeneous nucleation³⁰. The thermal diffusivity of o-terphenyl in the liquid phase is suppressed below the glass transition temperature, which likely promoted incorporation of embryos directly into the growing phase, rather than attachment of individual o-terphenyl molecules."

Current thinking on the cause of fast crystal growth from the glass [see Ediger and Yu, JPCB 119, 10124 (2015)] has focused on the failure of effective stress relaxation during the growth from the glassy state - an explanation that would make better connection with the observed size dependence and the large fluctuations observed by the authors.

We thank the reviewer for bringing the article JPCB 119, 10124 (2015) to our attention. We now cite this article as Ref. [32]. It is possible that the same explanation - effective stress relaxation does not occur, which promotes faster crystallization - could indeed apply to our experiments. However, currently we do not have a way to experimentally verify this. On the other hand, we have observed experimentally the presence of superclusters and how they couple to the growth front (Figure 4b). Therefore, we have direct evidence of superclusters that couple to the growth front, which enhances the growth rate.

In response, we have added the following discussion to the revised manuscript on page 9:

"We also note that the observed size dependence and fluctuations in the growth rate may be related to the potential lack of stress relaxation during growth from the glassy state, as previously studied in a polymer system³²."

The simulation results - that crystals grow slower when quenched fast than when quenched slow - really do not provide any strong support for the proposed cluster model. The simpler explanation is that the fictive temperature of the rapidly cooled liquid is higher than that of the more slowly cooled liquid and so the thermodynamic driving force for crystallization is less (another alternative explanation of the experimental observation).

We understand the reviewer's point that higher fictive temperature for rapidly cooled liquids could lead to slower crystallization due to a smaller thermodynamic driving force. However, we point out that the fast quenched liquids also show smaller fluctuations in the growth rate when heated compared to the slow quenched liquids (Fig. 3g). If the fictive temperature was the primary cause of enhanced crystallization, we would expect the growth fluctuations to be independent of the cooling rate. In addition, crystallization processes that involve incorporation of superclusters or embryos have been reported previously (ref [31]: PRB 91, 214103 (2015); ref [30] PRB 52, 3900 (1995)), conceptually in agreement with our observations. Therefore, we argue that the presence of superclusters influences crystal growth and faster quenching generates fewer superclusters.

In addition, the samples were studied at the isothermal crystallization temperature (360°C), which is much higher than the glass transition temperature (~ 230°C for the bulk), for times greater than tens of seconds. Thus, at these elevated temperatures, the memory of the fictive temperature would be largely erased, minimizing the role of the fictive temperature on growth kinetics. For the MD simulations also, the growth rate was measured once the Lennard-Jones glasses were heated to the crystallization temperature, again largely eliminating the effects of the fictive temperature.

In response to this comment, we have added the following sentences to the revised manuscript on page 8.

"For example, the crystallization rate can be affected by the fictive temperature. Generally, the fictive temperature is lower for slowly cooled liquids, which gives rise to a higher thermodynamic driving force for faster crystallization. However, the effects of the fictive temperature are only present near or below glass temperature. In our experiments, because the rods are crystallized at a temperature that is significantly higher than the glass temperature, the effects of the fictive temperature are negligible."

Some general comments:

i) Error bars should be included on the plot of the average crystallization rate in Fig. 3c.

We thank the reviewer for this suggestion. Because our images are resolved at the atomic scale with clear lattice fringes, we estimate that the uncertainty in measuring the growth rate is smaller than the width of one atomic plane layer, ~ 0.3 nm. Thus, we have added error bars (+/- 0.15 nm) to every data point in Figure 3c. We note that for the 65 nm rod with the red data points (heated glass), the error bars from the measurements are too small to be visible. The trend we observe is clearly outside of the error bars, and thus our interpretation stands. For Figure 3d, the error bars give the standard deviation of the growth rate. The revised Figures 3c and 3d with error bars are shown below.

In response to this comment, we have now included error bars in Fig. 3c as +/- 0.15 nm. In addition, we include error bars for the growth rate in Fig. 3d, given as the standard deviation of the growth rate measurements. In the revised manuscript, we include the following sentence on page 6:

"There is uncertainty in measuring the location of the growth front. Since our images are resolved at the atomic scale, the estimated uncertainty is +/- 0.15 nm."

In the caption to Fig. 3d in the revised manuscript, we have added, "The error bars represent the standard deviation of the growth rate."

ii) It appears that some references are missing - my copy only goes up to citation 37 and the paper cites papers up to 44.

In response to this comment, we fixed the citation numbering in the revised manuscript. There were a total of 38 references cited in the original manuscript, so ref. 44 should have been labeled ref. 38. The revised manuscript has a total of 41 references, since we added 3 new references, as suggested by reviewers.

Reviewer #3 (Remarks to the Author):

In this work, the authors investigated the crystal growth kinetics of Pt-based metallic glass nanorods using in situ TEM measurements. Specifically, the nanorods were first melted, then either rapidly quenched to crystallization temperature T_c , or to a much lower temperature and re-heated to T_c . The materials were then kept at T_c , at which isothermal crystallization occurred and the growth rate was recorded via in situ TEM. It was found that the re-heated material had crystal growth rate 26 times faster than the direct cooled sample. The authors attributed the difference to the direct attachment of metastable nanoclusters below size of critical nuclei, and supported the claim by comparison with MD simulation of a binary Lennard-Jones system. The carefully conducted in situ TEM measurements and the subsequent analysis convincingly demonstrated the difference in crystal growth speed from the liquid state versus the glass state, at least for the 65nm nanorods. The hypothesized cluster-coupled mechanism requires better scrutiny on the applicability/limitations for reasons listed below.

First, the authors presented their results in contrast to established theory where cooled sample contains more nuclei, which in turn will have higher nucleation rate but the same growth rate. However, the authors had identified that due to nano-confinement, only one nucleus grew (and formed a single crystal). By comparing the time scale involved: tens of seconds for the growth versus sub-microsecond for nucleation, the crystallization process of this material appears growth-limited.

We thank the reviewer for his/her insightful comment about nucleation- versus growth-limited crystallization. Like many others in the field, we have considered the nucleation part of the crystallization process, which is scientifically important and particularly complex during the early stages of nucleation. However, we are not able to determine the onset of nucleation due to the lack of spatial resolution, even though we are using an aberration-corrected TEM. We can identify the time when we observe a visible crystalline grain or a supercritical nucleus, which takes at least several seconds. As the reviewer states, studying nucleation is challenging yet worthy to pursue. However, we stress that this paper focuses only on growth.

In response to the reviewer's comment, we have added the following sentences to the revised manuscript on page 4.

"After reaching the crystallization temperature, it took at least several seconds before a stable nucleus was observed. The *in situ* TEM movies were acquired after the nucleation event to track the crystal growth kinetics. We define the start of the movie as $t = 0$ sec."

Without comparing with bulk material (for which the classic theory is based on) and analyze the density of nuclei, we cannot rule out with confidence that what's observed is merely a manifestation of an evolving super-critical nucleus engulfing sub-critical clusters. If one should expect multiple super-critical nuclei to form in 25 and 40 nm rods during the very rapid heating, then the argument of that we are observing something different from classic theory will be more convincing. As of now, it can be argued that the observation is size/material-system specific instead of having wide range implications.

We do not understand the reviewer's point here. What we report in our manuscript is that a super-critical nucleus (the single crystalline grain) engulfs sub-critical clusters, as the reviewer writes, *only* for crystallization initiated from the glass state and *not* for crystallization initiated from the liquid state. As a result, the growth rate is faster for the heated glass compared to the growth rate for the supercooled liquid at the same growth temperature. These results are in contrast to those from classical theory, which predicts the same growth rate irrespective of the thermal processing path as long as the growth temperature is the same. Because we focus on the growth rate, whether a single nucleation event occurred (resulting in one super-critical nucleus) or multiple super-critical nuclei formed is irrelevant.

Comparing our results to those from bulk materials is difficult because bulk Pt-based metallic glass would crystallize via solute-partitioning, which is dominated by long-range chemical gradients and diffusion, while our crystallization process is collision-limited since we observe a single-crystalline grain with a composition that is the same as the glass composition. Because the crystallization pathways are different between the bulk systems and nanorods, direct comparison of the growth kinetics is not possible. In addition, as most crystallization studies of bulk samples are *ex situ* and post mortem, direct measurement of the growth rate in bulk samples during crystallization is challenging.

In response to the reviewer's comment, we have added the following sentence to the revised manuscript on page 6.

"Comparison to growth kinetics of bulk samples is difficult because bulk samples crystallize via solute partitioning dominated by chemical diffusion fields, while the nanorods crystallize into a single crystalline grain, dominated by collision-limited kinetics."

Following the above argument, the appearing lack of difference in 25/40 nm material should be addressed. While the rate of 40 nm heated material as shown in fig 3(d) does appear higher, no error bar is given. Based on figure 3(c), the rate of cooled versus heated material seem very close factoring in fluctuations.

We agree with the reviewer that for small rods (the 23 nm diameter rod and even the 40 nm diameter rod), the growth rate begins to merge for the heated glass and cooled liquid. However, for the 65 nm diameter rod (Figure 3c) and for the 80 nm diameter rod (Figure 1), the growth rate asymmetry is clear. That is, the growth rate of a heated glass is much higher than that of a cooled liquid at the same temperature. Based on our hypothesized cluster-coupled growth model, the decreasing asymmetry of the growth rate for small rods can be explained by fewer subcritical clusters being available in small rods due to nanoscale confinement. This picture can consistently explain all of the observations in the manuscript.

In response to the reviewer's comments, error bars are now included in Fig. 3c. In addition, we include the standard deviation of the growth rate as error bars in Fig. 3d. We also added the following discussion to the revised manuscript on page 6.

"The vanishing asymmetry between the growth rates upon heating and cooling for small rods is interesting. With our current hypothesis, this behavior can be attributed to nanoscale confinement: fewer clusters are available for cluster-assisted crystallization for smaller nanorods. In this case, the growth

rate for the heated glass and cooled liquid do not differ significantly since there are only few clusters in both cases.”

Also, the authors have acknowledged that any size/population difference in sub-critical nuclei distribution should affect the rate. The authors claimed to test this (line 193), but only went on to address the appearance of ordered region instead of comparing growth speed.

This comment concerns Figure 4b. We expect that the stability and population difference in the sub-critical nuclei distribution will affect the growth rate, as the reviewer states. To investigate the distribution of subcritical nuclei experimentally, we varied the heating profile. Instead of rapidly heating the glass to 360°C, we slowly heated the glass to 340°C. The idea was that slow heating will induce more sub-critical nuclei and lower temperatures would make the nuclei more stable. We were able to directly visualize many nanoclusters in the glass region, preceding the growth front (as shown in Figure 4b). However, because the crystallization temperature is 20°C lower and also the nanorod has a different diameter, we do not directly compare the growth rate of Figure 4 to growth rates of Figure 3. Thus, Figure 4 serves to directly confirm presence of many subcritical clusters.

Figure 3b indirectly assesses the effect of the population difference in subcritical nuclei on the growth rate. The idea is that smaller rods will contain fewer subcritical clusters since they have a smaller volume. Indeed we see that for heated glasses the growth rate decreases with decreasing diameter of the nanorod, despite carrying out the experiments at the same crystallization temperature. For cooled liquids, the growth rate does not depend on the rod size, which is consistent with our hypothesis that the liquid does not contain many subcritical clusters.

Finally, the MD simulations showed cluster attachment can occur and will create significant spikes in growth rate curve. The simulation setup is for growth of a single crystal in a confined geometry, so again we have the problem of whether we are just observing size effect instead of new physics.

The MD simulations were performed using periodic boundary conditions in all three spatial dimensions, which mimics a bulk system. The boundary conditions of the simulations were not made clear in the original version of the manuscript.

In response to this comment, we have added the following sentence to the Methods section on page 13 of the revised manuscript:

“The MD simulations used periodic boundary conditions in all three spatial dimensions. Thus, the results from the MD simulations more closely mimic bulk samples, rather than those with nanoscale confinement.”

The simulation also claimed that material that underwent faster cooling also crystallized faster. This, however, is not addressed in the experiment.

First, the MD simulations find that faster cooling results in slower crystallization, which supports our hypothesis that faster cooling generates fewer clusters. We do agree with the reviewer that *experiments* that control the cooling rate and determine its effect on the growth rate are important. Due to the thermal drift during *in situ* TEM experiments, such controlled-cooling experiments are challenging. It is difficult to acquire TEM movies at atomic resolution while cooling rapidly due to thermal drift. In addition, controlling the cooling rate is difficult particularly in the cooling rate range we would like to explore. Thus, direct comparison between MD simulations and experiments is not currently possible.

Also figure 3(g) is a bit confusing. The left-hand axis plotting normalized standard deviations shows values of ~11 to 15. If it implies that the standard deviations are ~15 times the size of average speed, then any “difference” in the speed (about 20%) is meaningless.

The reviewer points out that, in Fig. 3g, the ratio σ_v/\bar{v} ranges from 10 to 14, which is large. σ_v and \bar{v} are the standard deviation and average values of the instantaneous growth velocities of the crystallization front. The reviewer suggests that a large value of σ_v/\bar{v} implies that the average speed can not be

measured accurately. In experiments, this would normally be true due to factors such as measurement errors, noise, and limited resolution of instruments. In other words, in experiments we do not know how true our measurements are due to imperfections, thus the standard deviation usually represents the degree of uncertainty with the measurements. However, simulations do not have these sources of error. Thus, in simulations, even though the standard deviation is large relative to the average, the measurement of the average can be obtained extremely accurately. The error in \bar{v} can be calculated as $\sigma_v / \sqrt{N_m}$, where N_m is the number of independent measurements of the instantaneous growth speed. We

carried out 16 independent simulation runs and 100 measurements per run, which is sufficient.

The standard deviation in our case represents the large jump in the growth front when a cluster attaches to the growing crystal. (If we omitted these cluster attachment events, the standard deviation will be very small.) We note that the system size will affect the magnitude of the standard deviation, i.e. as we increase the system size, σ_v / \bar{v} will decrease. For the small system we studied (the total number of atoms in the system was 2304 atoms with periodic boundary conditions), only one or a few clusters can join the growth front at each time, which can give rise to large fluctuations in the growth speed. By contrast, in a large system, there will be a continuous joining of many clusters at each time, which will give rise to smaller fluctuations. While the magnitude of the fluctuations would be affected by the system size, the trend, i.e. σ_v / \bar{v} scales with the cooling rate, holds true regardless of the system size because lower cooling rates give rise to enhanced local order.

In response to this comment, we include the following sentences (page 8).

“The relatively large standard deviation of the velocity compared to the mean velocity reflects that the system size in the simulations is small (2304 atoms), in which a single event of cluster attachment will appear dramatic. The large standard deviation does not mean that the average velocity is not accurately calculated in the MD simulations, which do not suffer from the measurement uncertainties that are found in experiments.”

Based on issues raised above, the authors should revise the manuscript before it can be published in high-impact journal such as Nature Communications.

We have fully addressed the comments by Reviewer #3, which have significantly improved the manuscript. We sincerely hope that the reviewer would find our response satisfactory.

Reviewers' comments:

Reviewer #1 (Remarks to the Author):

The authors have answered my questions. This paper is now acceptable for publication.

One question for the authors to think about, and make optional comment in the paper or elsewhere in future (like in a talk): recently Peter Harrowell published a Nature Materials paper reporting a fast crystal growth rate of something like 70 m/s for Ni from the melt. It was for collision limited growth without a noticeable energy barrier. I am not sure if the crystal growth in the current paper is also in that regime. Even if it is, in this case it is polymorphic but multicomponent, and the growth rate is drastically lower. But it would be interesting to compare these cases and comment on such differences.

Reviewer #3 (Remarks to the Author):

The authors have addressed most of my concerns in the initial review. However, there is confusion regarding a point I raised "Without comparing with bulk material ... size/material-system specific instead of having wide range implications." My question is should one expect to find multiple super-critical nuclei in ~ 65 nm (or less) of glass state material? Naturally growth will be faster if there is already a super-critical cluster in place, compared with having to form such cluster first. While super-critical clusters are obviously absent in the liquid state, if they occur infrequent enough that one is unlikely to find any in the 25/40 nm rods, then this can be a reason why the difference in crystallization speed vanishes for smaller rods. In other words, while naturally a growth-limited material, we need to rule out that size confinement does not limit the likelihood of having a super-critical nucleus from which subsequent growth proceeds.

I would also like the authors to clarify why for 40nm rods, the liquid state time scale is very different (fig 3c, middle panel, bottom axis). In the simplest case, this deviates from the commonly described methods of averaging over 200 TEM images to improve signal-to-noise ratio.

I believe the manuscript should be published once these concerns are addressed.

We thank the reviewers for the constructive comments. We provide a point-by-point response to their comments below. For clarity, the reviewers' comments appear in black and our responses appear in blue.

Reviewer #1 (Remarks to the Author):

The authors have answered my questions. This paper is now acceptable for publication.

One question for the authors to think about, and make optional comment in the paper or elsewhere in future (like in a talk): recently Peter Harrowell published a Nature Materials paper reporting a fast crystal growth rate of something like 70 m/s for Ni from the melt. It was for collision limited growth without a noticeable energy barrier. I am not sure if the crystal growth in the current paper is also in that regime. Even if it is, in this case it is polymorphic but multicomponent, and the growth rate is drastically lower. But it would be interesting to compare these cases and comment on such differences.

We thank the reviewer for pointing out this paper (Nat. Mater. 17, 881-886 (2018), "The mechanism of the ultrafast crystal growth of pure metals from their melts."). This paper discusses the origin of the barrier-less growth kinetics for pure metals using molecular dynamics simulations. We find this paper to be relevant to our work because the paper postulates that the ordering of the liquid at the solid/liquid interface contributes to the barrier-less growth. As the reviewer rightfully points out, our system is much more complex – multicomponent and polymorphic. However, the underlying hypothesis shares the same spirit, that is the local ordering of the supercooled liquid is coupled to the crystal growth.

In response, we have now added the following discussion (page 11) and cite the mentioned paper as Ref. [39].

"The local ordering of the supercooled liquid has been used to explain the fast growth kinetics of pure metals. For crystallization of pure metals from their melts, recent molecular dynamics simulations found that the liquid near the liquid / solid interface has crystalline ground states, which effectively leads to the absence of a barrier to crystallization for ultrafast crystal growth [39]."

Reviewer #3 (Remarks to the Author):

The authors have addressed most of my concerns in the initial review. However, there is confusion regarding a point I raised "Without comparing with bulk material ... size/material-system specific instead of having wide range implications." My question is should one expect to find multiple super-critical nuclei in ~65 nm (or less) of glass state material?

We thank the reviewer for the clarification. For the present manuscript, we focus on crystallization of a single super-critical nucleus in order to study the growth dynamics well after the nucleation event. Thus, for all rods (~25, 40, and 65 nm diameter), there is only one super-critical nucleus that grows. We are sure this is the case because the final crystalline phase is a single crystalline phase. Thus, the present study is about the difference in the population of sub-

critical clusters due to different processing conditions and how the population difference of sub-critical clusters affects the growth kinetics.

As the reviewer points out, multiple super-critical nuclei can certainly form in large metallic glass nanorods, which would then form poly-crystalline domains. In this case, the number of super-critical nuclei can be controlled by the diameter of the rods, purely by the geometrical confinement effect. We have studied this scenario in our previous paper, 'Tailoring crystallization phases in metallic glass nanorods via nucleus starvation,' Nat. Commun. 8:1980 (2017), cited as Ref. [15].

Naturally growth will be faster if there is already a super-critical cluster in place, compared with having to form such cluster first.

Yes, we agree with the reviewer – crystallization will be faster if a super-critical cluster is already in place. We stress that we are only looking at growth kinetics after the nucleation of the super-critical cluster. During growth, we do not observe any additional super-critical clusters forming in the liquid phase. Our hypothesis to explain the faster crystallization by heating the glass (rather than by cooling the liquid) is due to formation of sub-critical clusters, which can be found in the glass state, but absent in the liquid state.

While super-critical clusters are obviously absent in the liquid state, if they occur infrequent enough that one is unlikely to find any in the 25/40 nm rods, then this can be a reason why the difference in crystallization speed vanishes for smaller rods. In other words, while naturally a growth-limited material, we need to rule out that size confinement does not limit the likelihood of having a super-critical nucleus from which subsequent growth proceeds.

We attribute the vanishing asymmetry in the crystallization speed of smaller rods to the nanoscale confinement in which fewer subcritical clusters are available for cluster-assisted growth for small rods. The reviewer asks whether the number of super-critical clusters would also be affected by the size confinement. We do not think so because our studies are limited to crystallization out of a single super-critical cluster. A super-critical cluster would be a thermodynamically stable cluster that would grow. In our studies, we have only one cluster that grows, with the final state being single-crystalline domain.

I would also like the authors to clarify why for 40nm rods, the liquid state time scale is very different (fig 3c, middle panel, bottom axis). In the simplest case, this deviates from the commonly described methods of averaging over 200 TEM images to improve signal-to-noise ratio.

The different time scale for the cooling experiment of the 40 nm rod (Fig. 3c, middle panel, bottom axis) is due to the particular experimental conditions. The movie was taken at a higher magnification with a small field of view, which meant that we could not track the growth for a long time. For this particular rod, we only had a short segment of the TEM movie that could be analyzed, resulting in the different time scale. For this dataset, we did not average over 200 TEM images.

In response, we clarify this point in the figure caption of Figure 3.

“For the cooling experiment of the 40 nm rod, the growth dynamics was tracked for only a short time. In this case, the images were not averaged over 200 frames.”

I believe the manuscript should be published once these concerns are addressed.

We thank the reviewer for carefully reviewing our manuscript, which represents a considerable amount of time. We hope that our responses address the reviewer’s concerns and clarify the confusion on the super-critical nuclei questions.

REVIEWERS' COMMENTS:

Reviewer #3 (Remarks to the Author):

The authors have satisfactorily addressed my concerns in the previous reviews. Thus I believed the manuscript should be published.

REVIEWERS' COMMENTS:

Reviewer #3 (Remarks to the Author):

The authors have satisfactorily addressed my concerns in the previous reviews. Thus I believed the manuscript should be published.

We thank the reviewer for spending his/her valuable time on reviewing our manuscript.